# I Need to Know: Using the CeHRes Roadmap to Develop a Treatment Feedback Tool for Youngsters with Mental Health Problems

**DOI:** 10.3390/ijerph191710834

**Published:** 2022-08-31

**Authors:** Ilja L. Bongers, David C. Buitenweg, Romy E. F. M. van Kuijk, Chijs van Nieuwenhuizen

**Affiliations:** 1Institute for Mental Health Care Eindhoven (GGzE), Centre for Child and Adolescent Psychiatry, P.O. Box 909, 5626 ND Eindhoven, The Netherlands; 2Scientific Center for Care & Wellbeing (Tranzo), Tilburg University, P.O. Box 90153, 5000 LE Tilburg, The Netherlands

**Keywords:** co-design, youngsters, digital treatment feedback tool, CeHRes roadmap, e-mental health

## Abstract

Patient-Reported Outcome Measures (PROMs) are often used to monitor treatment outcomes in youth mental health care. Unfortunately, youngsters are rarely informed about the results of their PROMs or, when they are, it is in an insufficient manner. Therefore, a web application was developed—together with youngsters—aimed at giving them feedback about their PROMs. The aim of this study is to describe the development process of the application. An expert panel consisting of youngsters, web designers and researchers, as well as a representative from a client organisation, developed the e-health application INK (short for ‘I Need to Know’) in an iterative process based on the Centre for eHealth Research roadmap (CeHRes roadmap). Youngsters prefer, among other aspects, a simple, easy-to-use e-health application with a colourful appearance and want to be able to compare their results across different time points and informants. The INK tool provides youngsters with insight into their PROM results. Based on the youngsters’ preferences, INK users can choose which feedback information is visible. INK facilitates youngsters’ active participation in their treatment as well as shared decision-making with their professional caregivers.

## 1. Introduction

Providing feedback to youngsters in youth mental health services is of utmost importance [1,2,3,4]. Feedback has a consistently positive effect on youngsters during their treatment [5]. It has, for instance, a positive effect on youngsters’ well being [4] and a small effect on dropout reduction [1]. Including regular outcome feedback in youth mental healthcare helps prevent and attenuate the potentially severe long-term effects of mental health problems [6]. Taking into consideration the numerous advantages of providing outcome feedback to youngsters in youth mental health services, it is surprising that this feedback is not always provided [7,8,9]. Particularly in light of the new opportunities that digital technologies provide. E-mental health applications have the flexibility to tailor content in a personalized way [10,11] and engage users in real-time [12]. Both personalisation and real-time engagement have been found to improve responsiveness and patient motivation [13]. Moreover, e-mental health applications seamlessly fit in with everyday experiences and allow youngsters to take the lead [14,15,16]. E-mental health is, therefore, especially suited to providing feedback to youth in mental healthcare.

The importance of co-design and the active involvement of youngsters and other stakeholders across the entire design process of an e-mental health application or intervention has been highlighted in several studies [17,18,19]. Co-design is more than the consultation of stakeholders; it involves exploring and articulating needs and developing solutions together with them. In this way, co-design results in a richer and deeper understanding of the end-users. The Centre for eHealth Research roadmap (CeHRes roadmap; [20]) provides a framework for involving end-users in the development process of e-mental health applications through co-design. Co-design has been identified as an effective design philosophy for guiding the development of e-mental health applications for use in youth mental healthcare [17,19]. The benefits of co-design are well established: it enhances the range of available design ideas [19], increases the understanding of user needs [17] and improves the support and success of the application by focusing on user needs [18,19].

As of yet, little work has been performed to identify youngsters’ preferences regarding the design and functionality of a digital Patient-Reported Outcome Measurement (PROM) feedback tool. A recent qualitative study by Mayworm and colleagues [21] revealed that easy-to-use, personalisation and feedback are important preferences for middle and high school students. It is unclear, however, whether these preferences remain when explored in the context of the development of an e-mental health PROM feedback tool for youngsters with mental health problems. Therefore, the aim of this study involves the co-design-driven development of a digital PROM feedback tool for use in youth mental health care. Youngsters’ preferences regarding the design and functionality of the tool receive special attention.

## 2. Methods

### 2.1. Expert Panel

An expert panel was involved throughout the process of the design and development of a digital feedback tool INK (short for ‘I Need to Know’). As the literature emphasises the importance of dialogue between designers, end-users and other important stakeholders [20], the panel consisted of youngsters, web designers with a focus on user experience and a researcher, as well as a representative from a client organisation. Seven youngsters aged 12–18 were invited to participate in the panel, and all of them were required to have (previous) experience with youth mental healthcare and to have been treated in a mental health care institute for psychiatric problems. The representative of the client organisation recruited the youngsters. The client organisation is a network of client advisory boards in care and wellbeing (www.loc.nl) (accessed on 27 July 2022). A minimum of three youngsters was required to proceed with individual panel meetings. Every panel meeting was attended by three to seven youngsters (mean attendance: 5.4 youngsters), two user experience web designers, a researcher and a representative from a client organisation. Panel meetings took place in an informal setting in the region where the youngsters were living on Wednesdays from 17.00–18.30. Various actions were undertaken to minimize power structures that typically exist between adults and children, such as engaging in informal activities to get to know each other, using informal language, wearing casual outfits and eating snacks and dinner together. After each expert panel meeting, the researchers and designers discussed the meeting and prepared the subsequent design steps and the next panel meeting. An iterative process was established as panel meetings were based on the information gathered in the previous panel meetings. The meetings were not recorded, but the researcher took extensive notes and wrote a report of the meetings. At the start of each panel meeting, the general outline of the meeting was shared with all the youngsters. Next, a recap of the previous meeting was given, and the report of the previous panel meeting was validated.

### 2.2. Development Process

The iterative co-design-driven development of the digital feedback tool INK was structured according to the first three phases of the CeHRes roadmap: (1) Contextual Inquiry, (2) Value Specification and (3) Design [17,20]. In practice, these three phases were not executed sequentially but were interwoven throughout the developmental process. The CeHRes roadmap is based on various evidence-based models and frameworks, such as human-centred design and participatory development [20]. The three phases of the CeHRes roadmap are schematically displayed in Figure 1. Below, the phases and how they have been adapted to fit the current study are described in more detail. To triangulate data and develop a better understanding of youngsters’ needs, the CeHRes roadmap involved a formative evaluation of the results gathered during each phase.

#### 2.2.1. Phase 1: Contextual Inquiry

The main goal in this first phase was to achieve an understanding of future users and the current situation. In the current study, the Contextual Inquiry was initiated by identifying stakeholders and establishing an expert panel to represent future users. In addition, to provide youngsters with insight into the current situation of the PROM feedback that was offered a patient journey was incorporated (Table 1). This is an interactive method to allow youngsters to become more familiar with the research project and the goal of the co-design-driven development of INK [22].

#### 2.2.2. Phase 2: Value Specification

The Value Specification phase served to clarify what values were important to future users and how these values may be translated into user requirements. In the current project, the youngsters were asked what functionalities they would like to see in INK. In addition, they were invited to share factors that would prevent them from using INK or that would encourage them to employ the tool. Furthermore, youngsters were invited to share their preferences regarding the design of INK. Based on this input, the relevant values were formulated.

#### 2.2.3. Phase 3: Design

The Design phase started with a series of sketching exercises by the youngsters. First, they were invited to sketch as many possible visualisations of questionnaire results as possible using pencil and paper. Next, they were asked to draw possible depictions of (1) the distinction between high and low scores, (2) the results of assessments at different time points and (3) the outcome of questionnaires answered by different informants. Based on these images, the designers constructed five possible directions for the design (paper prototypes). The panel discussed these alternative directions and the preferred one was explored further using digital mock-ups. Different colour schemes were compared and discussed by the panel. Next, the designers developed a low-fidelity prototype that was examined extensively with the panel. The youngsters were invited to test the prototypes using either an Apple iPad 2 or Apple iPad 2017 provided by the researcher or their personal smartphones. Based on their feedback, the prototype was adjusted and refined where necessary, resulting in a final version of INK.

#### 2.2.4. Formative Evaluation

In this study, a mixed methods design was employed for the formative evaluation to triangulate the data and gain a better understanding of youngsters’ needs. The needs identified in the panel meetings were categorized into values, and these were translated into design options in INK. Throughout the development of INK, youngsters were invited to evaluate the current sketch, design or prototype using various methods, and after each evaluation, the results were discussed with the panel. During the initial steps of the development, the youngsters were invited to use four different stickers to assess ideas for the design and functionalities of INK (see Figure 2). Later, they used sticky notes to express their opinions about the sketches and prototypes. Moreover, decisions were sometimes made democratically by choosing the option that received the highest number of votes from the panel.

Near the end of the development of INK, the youngsters were invited to evaluate the degree to which their preferences were reflected in the tool. Usability tests were used to observe how youngsters navigate through the application and to see whether they were able to complete typical tasks. The interpretation of feedback circles was tested using a structured interview. They were shown four different feedback circle options. The one that was most often interpreted in a correct way was chosen for the final version of INK. The layout and colour scheme were tested using a questionnaire with a 5-point Likert scale.

### 2.3. Ethical Approval

This study was conducted according to the guidelines of the Declaration of Helsinki, and approved by the Ethics Committee of the Tilburg School of Behavioural and Social Sciences at Tilburg University (EC-2017.78). Informed consent was obtained from each participant. The youngsters received 25 EUR and a reimbursement of travel expenses via bank transfer for each panel meeting they attended. Occasionally, they were given small gifts such as a cinema voucher or card game during the course of the development process. Personal details were collected with the participants’ consent to make bank transfers possible.

## 3. Results

The co-design-driven development process consisted of ten sessions with the expert panel. The results were structured according to the three phases of the CeHRes roadmap: Contextual Inquiry, Value Specification and Design. The results of each phase were visualized with sketches and screenshots of the low- and high-fidelity prototypes. During the Contextual Inquiry (Phase 1), the trajectory of a typical treatment of a youngster was described using a patient journey. Important values (Phase 2) for future users were identified, and their translation into user requirements was discussed in the Value Specification phase. In the Design phase (Phase 3), the values were translated into a prototype of the digital tool INK.

### 3.1. Phase 1: Contextual Inquiry

During the Contextual Inquiry (Phase 1), the trajectory of a typical treatment of youngsters receiving mental healthcare was described using a patient journey. At the start of treatment, youngsters, their parents and sometimes their teachers were asked to fill out questionnaires in a typical treatment trajectory. In the panel meeting, the youngsters discussed the order of events in a typical treatment trajectory. For instance, whether they would prefer to complete the questionnaire before attending the mental healthcare institute for the first time or after. The youngsters in the panel meeting said that they only rarely received feedback about the questionnaires they filled out. They would, however, like to be given feedback, preferably directly after they finished the questionnaires.

### 3.2. Phase 2: Value Specification

An overview of all the youngsters’ preferences and needs regarding the functionality of INK is provided in Table 2. The needs were translated into design requirements in INK. Not all needs initially mentioned by youngsters were incorporated into the final version of the INK application due to technical and organisational constraints. Needs that were not part of the final version are denoted with an asterisk in Table 2. As can be seen in Table 2, the needs were categorized based on five values: Being well-informed, Personalized user experience, User friendly, Being in control and Privacy and safety.

#### 3.2.1. Phase 2 Value 1: Being Well-Informed

During the expert panel, the youngsters emphasized that they want to see how their treatment progresses in one screenshot. They also mentioned the importance of being well-informed about specific behaviours and not only receiving general feedback about their treatment progress. They indicated that it was important to view results from several time points and multiple informants in a single overview. The youngsters used a ‘lock’-sticker to indicate that this was essential for the INK tool (see Figure 3).

All needs related to the value of Being well-informed were translated into the final design of INK, and the way in which feedback should be visualized, in particular, was thoroughly discussed with the youngsters during the panel meetings. The general feedback was visualized using the colours green, orange and red to represent normal, subclinical and clinical scores, respectively. Besides the general feedback, specific feedback was added to the INK tool. By clicking on the green, orange or red feedback circle, a feedback score appears in numbers (range 0–100). According to the youngsters, these so-called ‘traffic light’ colours are familiar and likely to evoke similar associations among people (see Figure 4). However, they emphasized that these colours should be used modestly. It can be somewhat startling to see that your own scores are green and those for other informants show red or to see that your previous scores were green but new ones are red. At the same time, the youngsters did find it important not to sugar-coat negative scores: ‘it is how it is’.

#### 3.2.2. Phase 2 Value 2: Personalized User Experience

The youngsters emphasized the importance of the personalisation of their user experience. One suggestion was to create a profile page where youngsters can upload a personal picture and list some information, such as their name, age and personal interests. Later on, in the development process, this idea was replaced by using icons and avatars that are engaging, fun and age-appropriate (12–18 years), according to the expert panel. This resulted in functionality in which youngsters can ‘build’ their own avatar. Within the avatar builder, users can select their preferred gender and clothing as well as skin and hair colour (Figure 5). Youngsters commented that the avatar builder should not be too extensive, as the primary goal of the INK application was not to provide a game-like experience but to provide treatment feedback: ‘It is fun that you’re able to choose your own character, but keep it simple’. Another feature that enhances the personalisation of the experience was the matching of the avatar’s facial expressions with both the topic of the questionnaire and the displayed scores (see Figure 5). Lastly, there is an option for users to select their colour of preference for the general layout of the INK application.

#### 3.2.3. Phase 2 Value 3: User Friendly

Youngsters mentioned that a cheerful, non-depressing appearance of the tool motivated them to use INK, whereas the colour red, as well as sad or avoidant avatars, demotivated them (see Figure 6). The youngsters emphasized that INK should be applicable to all youngsters, so the language used in INK was adapted to the reading level of youngsters, and special attention was paid to avoiding stigmatizing language. For instance, the term ‘emotion and behaviour’ is used instead of ‘complaints’. During the panel meeting, special attention was paid to the usability of INK, resulting in an intuitive structure that makes INK easy to use. Only a few clicks are required to navigate through INK.

#### 3.2.4. Phase 2 Value 4: Being in Control

Youngsters stated that it is important that they are in control during their treatment. They emphasized being able to take control when they receive feedback about their treatment progress. The benefits of being informed were discussed in the panel meeting, and the youngsters mentioned that INK could also facilitate shared decision-making. Several features of INK give the youngsters control over the feedback they want to receive. For example, users are able to select or hide the informant or the scale scores based on what they want to see. INK has a predefined email format, which the youngster can use to ask for help (see Figure 7). All the options and functionalities of INK are explained in a short, clear tutorial that is always accessible to users. Not all preferences expressed by the youngsters that related to the value of Being in control could be implemented in INK. For instance, the youngsters suggested a chat option with their therapists and peers or an option to indicate when their opinions differ from their parents or teachers to facilitate discussions with their clinicians. These options could not be implemented because of technical constraints.

#### 3.2.5. Phase 2 Value 5: Privacy and Safety

The youngsters highlighted the importance of receiving treatment feedback in a safe way and that their activities in INK should not be followed by any organisation. Tracking and personalized advertisements abolished the feelings of INK being safe to use. Therefore, the youngsters in the expert panel were strongly opposed to the tracking of their use of INK. For safety and privacy purposes, only youngsters themselves can use the INK application, using their individual email address provided through the electronic patient file of the mental health institute and a password (see Figure 8). In this way, the safety and privacy of the information shared in INK are assured, and user information collected by INK will not be stored.

### 3.3. Phase 3: Design

Using paper and pencils, youngsters visualized the different questions or scales of the PROM questionnaires. Most sketches focused on faces and facial expressions (see Figure 9).

The initial suggestions, sketches by youngsters and identified values were used to create the first paper or low-fidelity prototypes of INK (see Figure 3 and Figure 4, for example). During the iterative development process, the list of needs regarding the design of INK expanded. The INK application should have a minimalistic, simple design. Although the questionnaires’ topics can be quite tense, youngsters clearly stated that the INK application should have a cheerful, non-depressing appearance: ‘Seeing a happy face can give a sense of relief, whereas a sad face can also make you feel sad’. According to youngsters, the design should be colourful but not too vibrant in order to maintain a simplistic design. However, most youngsters preferred pastel shades, some preferred bright, cheerful colours for the design of INK. Therefore, youngsters are able to select their preferred colour in the INK application. Based on this developmental process, the first prototype was designed. The usability of the prototype was tested with six youngsters of the expert panel using a structured interview; they all interpreted the feedback circles correctly. The layout and colour scheme were tested, and the youngsters were not satisfied with the colours of INK (see Figure 3, Figure 4 and Figure 7). Therefore, the colours changed from blue in the high-fidelity prototype to purple in the final version. See Figure 10 for the final version of INK that is implemented in a mental healthcare institute for child and adolescent psychiatry.

## 4. Discussion

This study involved the co-design-driven development of INK, a digital PROM-feedback tool for use in youth mental healthcare. Sharing feedback with youngsters in treatment is ubiquitous, but INK is unique. Although there are feedback tools for adults [2,23] or feedback tools that help clinicians to share feedback with clients [24], feedback tools developed especially for young people in mental healthcare settings in co-creation with youngsters were not available up to now. The contribution by the youngsters in the expert panel resulted in a final version of INK that enables youngsters to view the results of their PROMs in an accessible, appealing and intuitive way. The design requirements articulated by the expert panel related to the values Being well-informed and Privacy have all been implemented in INK. Only some of the requirements related to the value of Being in control could not be implemented in the design due to technical and organisational constraints. Whereas the values involving being User friendly and Personalized user experiences were important for the preferences of the youngsters according to the design and layout of INK, the values Being well-informed in addition to Privacy cover the main goal of INK.

The five values and corresponding user requirements identified in this study are comparable to the ones identified in the literature [4,25]. Being well-informed when receiving treatment feedback is an important value for youngsters. According to the youngsters in the panel, feedback is most useful when it involves insight into changes over time, is provided by multiple informants, includes additional information on questionnaires, scales or items and is comprehensive and well-organized. The importance of this value is in line with previous studies [12,26,27,28,29]. Feedback becomes more valuable for youngsters when it provides them with sufficient information to accurately interpret their PROM results. Although feedback does not directly increase the effectiveness of treatment [7], feedback can support shared decision-making, which increases compliance with the recommended treatment [30]. Hence, the use of INK to complement treatment may indirectly improve the effect of treatment.

Personalisation is frequently cited as one of the important advantages of e-mental health applications and is also related to increased engagement [31,32]. A tool that offers personalisation to the interests and needs of the user has a greater capability of persuasion and increases engagement with and adherence to the application [33]. It is, therefore, not surprising that the youngsters indicated the importance of age-appropriate content and a customizable avatar and background colour scheme. Similar preferences have been identified in studies with children and adolescents by Beck and colleagues [12] and Hetrick and colleagues [27]. In a study by Buitenweg and colleagues [26], however, young adult and adult participants explicitly rejected customizable avatars or colours. This difference between youngsters and adults implies that personalisation and user friendliness is age dependent, so it is important to identify relevant end-users and embrace them in the development process [17].

The youngsters in the panel emphasized the importance of the user friendliness of e-mental health applications in general and digital PROM-feedback in particular. Similar values have been identified by youngsters in virtually all comparable studies [12,34]. The way youngsters interact with a digital tool influences their engagement as well as the extent to which they revisit the tool [35,36]. This finding confirms the importance of ensuring that users find it easy to interact with and navigate through an e-mental health application.

The youngsters extensively indicated their need to be in control when using a digital PROM-feedback tool as well as the secure storage of their data (the Privacy and safety value). Having control over data access was highly valued by youngsters. The ability to notify other individuals and to have control over data access echoes findings by several other works [12,26,37]. It is important to note that youngsters’ preference for having control over data access may conflict with clinicians’ wish to have unrestricted access to youngsters’ PROM results. In addition, studies indicate that feedback is most effective when provided to both youngsters and clinicians [2,5]. Maathuis and colleagues [38] identified this value conflict during the development of a web-based quality of life assessment instrument for youngsters and adults with mental health problems. These results emphasize that there exists a potential conflict between youngsters’ need for autonomy and clinicians’ wish to have unrestricted access, which may influence the effectiveness of feedback. The user requirements identified in this study may inform the future development of digital feedback tools for youngsters with mental health problems. Adhering to youngsters’ preferences whenever possible may improve the often-problematic implementation of treatment feedback for youngsters with mental health problems [39].

## 5. Strengths and Limitations

The involvement of both youngsters and web designers in the panel is an important strength of this study. As Mulvale and colleagues [18] and Thabrew and colleagues [19] indicated, it is vital that web designers come into contact with the end users throughout the development of an e-mental health application. The consistency of the composition of the panel is another important strength. The youngsters were involved in multiple panel meetings (up to ten meetings). This enabled them to provide feedback on the current version of INK and to monitor to what degree their feedback was processed in subsequent versions. At the same time, three important limitations should be taken into account when interpreting the results of this study. The first limitation pertains to the size of the panel as well as the sampling strategy employed to recruit it. The youngsters joined the panel based on a convenience sampling strategy. The results should, therefore, be generalized with caution. In addition, the number of youngsters who contributed to the panel was limited, and little information regarding their background was available. Some panel meetings involved only three or four youngsters. The small size of the panel, however, also had two important benefits. It facilitated collaboration and resulted in an atmosphere of trust in which the youngsters could share their experiences and opinions. The absence of care professionals in the panel forms a second limitation. The professional perspective has been identified as very important in the development of e-mental health applications [40,41]. The preferences described in this study were not identified in systematic experiments but during an organic development process, which forms the third limitation. The ecological validity of the results, however, may benefit from this.

## 6. Future Research

This study provides a number of important opportunities for future research. First, it is crucial to assess to what degree the use of INK by youngsters in their treatment improves their treatment satisfaction and symptom levels or other clinical outcomes. Second, research into the optimal implementation strategies for INK is essential. As the literature reveals and experience teaches, many digital mental health tools fail to have an impact on the lives of youngsters and clinical practice. Frameworks such as the Consolidated framework for implementation research [42] or Proctor’s framework of implementation outcomes [43] may be helpful when studying the implementation of INK [44]. Improving the flexibility of INK so that it may be used in multiple care settings and by various groups, such as parents, is a third important avenue for future research. Currently, INK includes a limited number of questionnaires. By adding further ones, INK would become useful for additional patient groups.

## 7. Conclusions

An easy-to-use tool for treatment feedback has been developed, together with youngsters. The five values of Being well-informed, Personalized user experience, User friendly, Being in control and Privacy and safety were identified during the development and have been incorporated into the final version of the tool. INK enables youngsters to actively participate in their treatment and to make informed decisions together with their professional caregivers.

## Figures and Tables

**Figure 1 ijerph-19-10834-f001:**
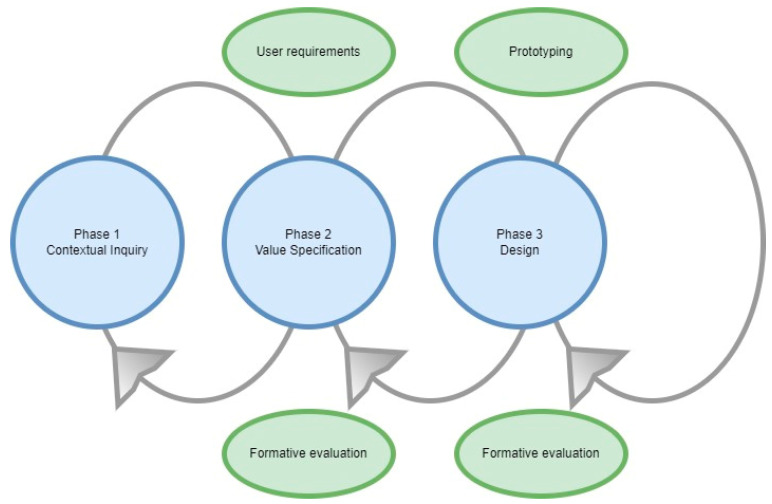
Phases of the CeHRes Roadmap.

**Figure 2 ijerph-19-10834-f002:**
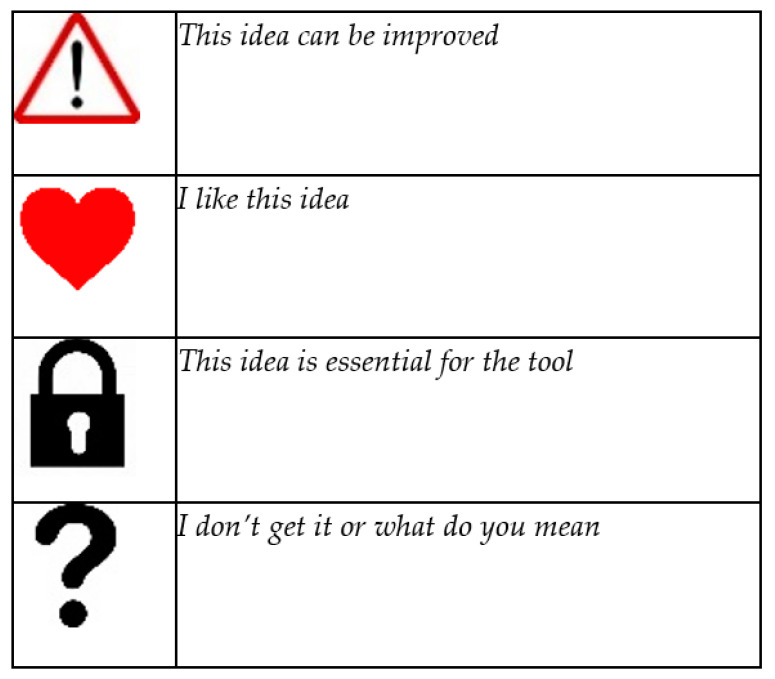
Stickers used by the expert panel.

**Figure 3 ijerph-19-10834-f003:**
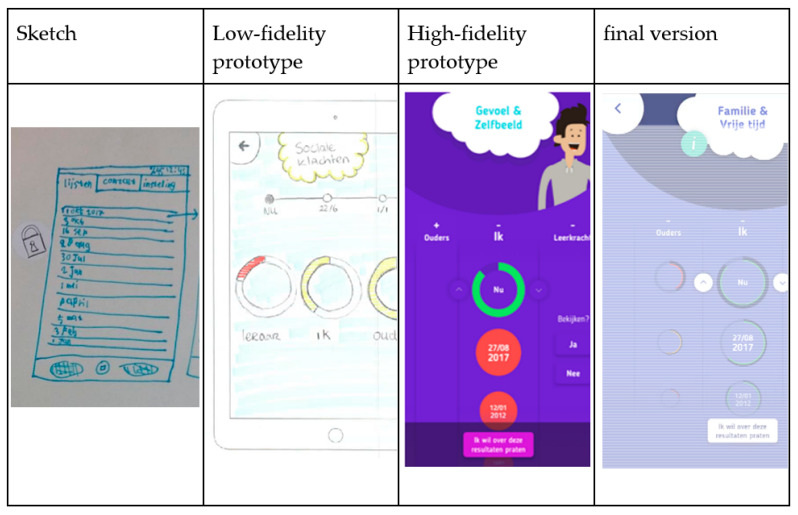
Youngsters indicated that a well-organized application was essential.

**Figure 4 ijerph-19-10834-f004:**
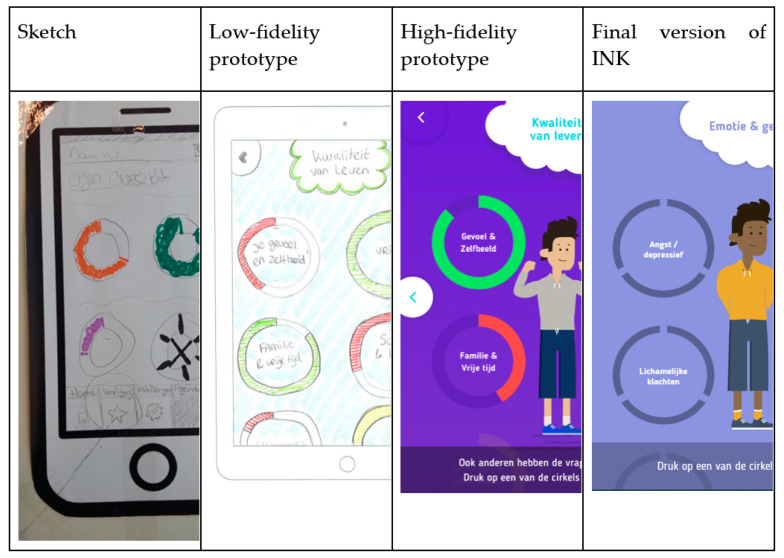
Use of traffic light colours in feedback.

**Figure 5 ijerph-19-10834-f005:**
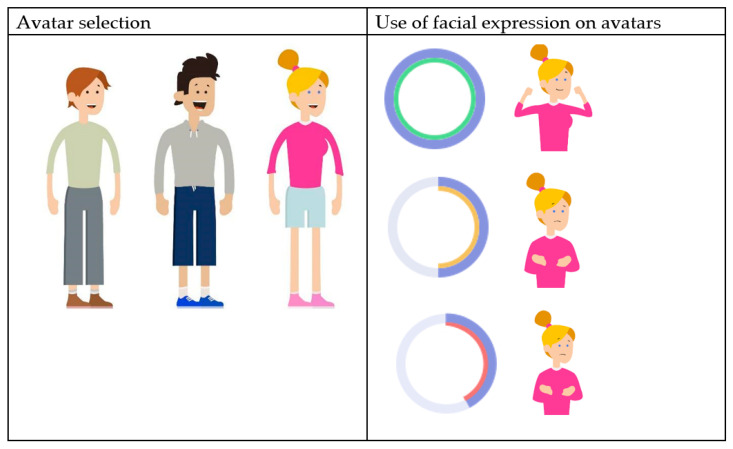
Use of avatars in INK.

**Figure 6 ijerph-19-10834-f006:**
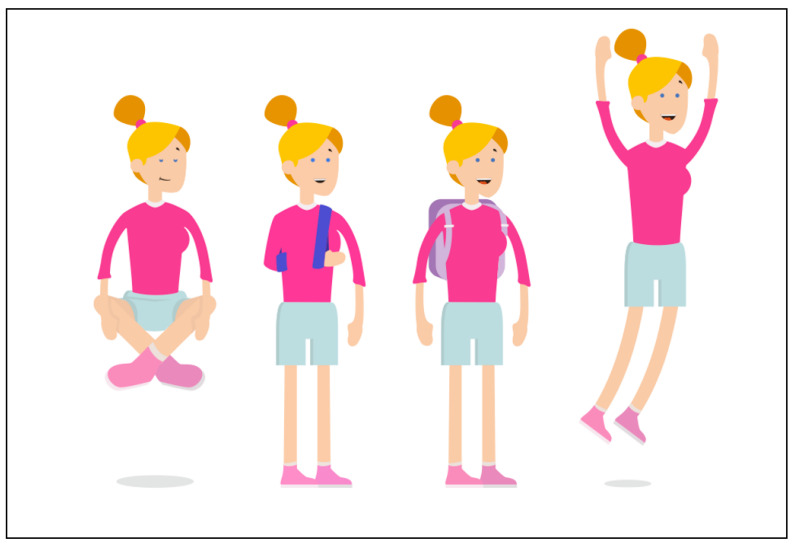
Bright and cheerful avatars on the specific behaviour page.

**Figure 7 ijerph-19-10834-f007:**
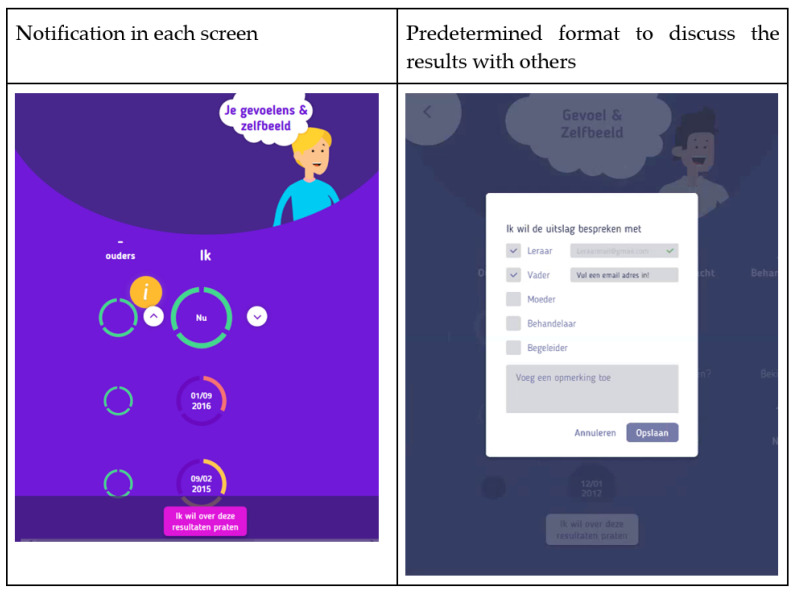
Notification ‘I want to discuss these results with others’.

**Figure 8 ijerph-19-10834-f008:**
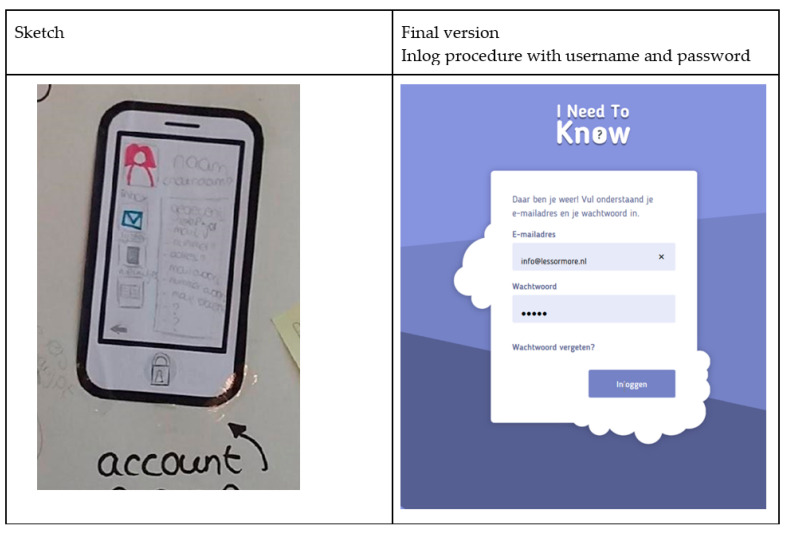
Privacy and Safety: Accounts are only accessible by youngsters.

**Figure 9 ijerph-19-10834-f009:**
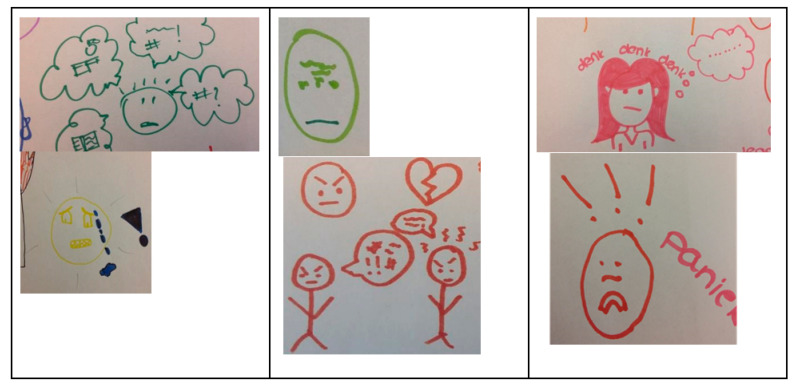
Visualisation questionnaires.

**Figure 10 ijerph-19-10834-f010:**
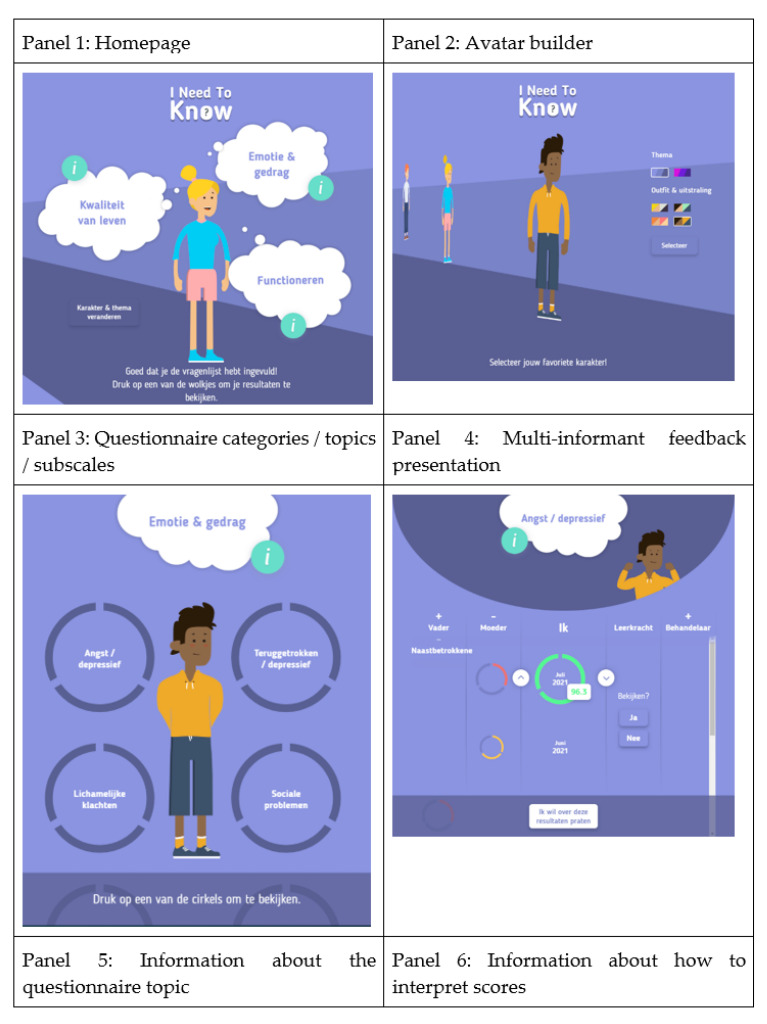
Final version of the INK tool.

**Table 1 ijerph-19-10834-t001:** Tools and techniques used during the development of INK.

CeHRes Phases	Goals	Activities
Contextual Inquiry	Identify stakeholders	Compose expert panel
	Investigate the strong and weak points	Patient journey
Value Specification	Identify values of stakeholders	Brainstorm
	Prioritize values	Use stickers and smileys
	Technology requirements	Sticky notes and votes
Design	Introduction of questionnaires	Sketches by youngsters
	Low-fidelity prototypes	Usability test
	High-fidelity prototypes	Structured interview

**Table 2 ijerph-19-10834-t002:** Overview of preferences of the youngsters for the design of INK-Value Specification.

Value	Needs	Translation of Needs in INK
Being well-informed	Track feedback over time	The INK application enables users to keep track of their treatment feedback over time.
Being well-informed	Multi-informant feedback	The INK application allows users to select and compare treatment feedback from multiple informants.
Being well-informed	Comprehensive and well-organized feedback	Take in everything at a single glance. The INK application shows all feedback related to one (subscale of a) questionnaire at a glance.
Being well-informed	General and specific feedback	The INK application provides both general (normal, subclinical and clinical) and specific scores (0–100).
Being well-informed	General feedback	The colours green, orange and red are used modestly to represent normal, subclinical and clinical scores, respectively.
Being well-informed	Specific feedback	The INK application provides insight into the treatment progress within a specific score on a subscale of a questionnaire (0–100).
Being well-informed	Information about (subscales of) questionnaires	The INK application informs users about the meaning of (subscales of) questionnaires in easy-to-understand language. Mentioning positive and negative examples.
Personalized user experience	Profile page *	Profile page, including the possibility to upload a picture and personal information such as name, age and personal interests.
Personalized user experience	Tailoring	Youngsters expressed different ways to adapt the application to youngsters’ characteristics. For example: showing different icons based on age, as well as allowing users to ‘build’ their own character and upload their own picture.
Personalized user experience	Avatar	The avatar in INK shows personality and facial expressions.
Personalized user experience	Content tailoring	The avatar in INK shows different emotions based on the topic of the questionnaire and the score.
Personalized user experience	Age-appropriate	The design and user experience of the INK application should be age-appropriate (12–18 years).
Personalized user experience	Minimalistic design	The INK application has a minimalistic, simple design, which is colourful but not too vibrant in order to maintain a simplistic design. Users are able to select their colour scheme of preference in the INK application: pastel shades or bright, cheerful colours.
User friendly	Cheerful design	The INK application has a cheerful, non-depressing appearance. Predominant use of the colour red and avoidance of sad facial expressions.
User friendly	Age-appropriate	Names and descriptions of questionnaires are adapted to the reading level of youngsters.
User friendly	Positive affirmation	The INK application rewards end-users for filling out the questionnaires by means of positive affirmations (e.g., ’Thanks for filling out the questionnaire’ or ‘Good job!’)
User friendly	Easy to use	The INK application is easy to use.
User friendly	Easy to navigate/speed of use	Users are able to reach the desired questionnaire results in only a few clicks.
User friendly	Well-functioning	The INK application can be used on a mobile phone, tablet or computer.
Being in control	Express agreement *	A desirable feature is to report whether you agree with the reported questionnaire results. For example, when youngsters observe that their parents, teachers and/or clinicians hold different opinions about the youngsters’ behavioural and emotional problems, it is nice to discuss these different opinions.
Being in control	Notification to other informants	Another desirable feature is to mark specific questionnaire results you would like to discuss with others, such as one’s relatives or therapist.
Being in control	List of marked questions *	‘Marked’ questions should be saved on a list, so they are easy to retrieve. This list can serve as a conversation aid in one’s therapy session.
Being in control	Reminder *	Reminder when a new questionnaire is available to fill out and/or when there are new results available.
Being in control	Chat *	The INK application should have a chat function to ask feedback-related questions to your therapist and/or (anonymous) contact with peers to share feelings.
Being in control	Calendar *	The INK application should have a calendar displaying all (treatment-related) appointments and tasks.
Being in control	Tutorial	The INK application shows a pop-up with short instructions on first use (tutorial).
Being in control	Homepage	Attractive homepage and an indication of the application’s main features.
Being in control	Self-management	The youngsters decide who can access their questionnaire results.
Being in control	Data access	Only youngsters have access to INK.
Privacy and safety	Safe storage of data	Data should be handled confidentially and stored in a safe manner.
Privacy and safety	Advertisements	The INK application is free of (personalized) advertisements.

*: Needs are not added to the final version of INK.

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
