# Peer review of "I Need to Know: Using the CeHRes Roadmap to Develop a Treatment Feedback Tool for Youngsters with Mental Health Problems"

_ijerph, 2022, doi:10.3390/ijerph191710834_

Round 1
Reviewer 1 Report
Dear authors,
First of all, thank you for submitting the manuscript entitled: “I Need to Know: Using the CeHRes roadmap to develop a treatment feedback tool for youngsters with mental health problems” for consideration for publication in the International Journal of Environmental Research and Public Health. According to the information of The Special Issue “The New Normal in Mental Health Care”, authors are invited to submit papers which combine a high academic standard with a practical focus on providing insight into why, how, when, and by whom mental health care can be brought to its new normal. Therefore, the paper is suitable for this Special Issue since it aims to develop a digital PROM feedback tool for use in youth mental health care, a very relevant topic.
The paper is well written and I find the manuscript interesting and very significant in the area, since the use computer technologies, particularly web- applications may help improving the use of self-reported measures, and facilitating youngsters’ active participation in their treatment.
One of the major positive points was the contribution by youngsters in the expert panel, since it might improve the problematical implementation of treatment feedback for youngsters with mental health problems.
As mentioned by the authors, the sample size is a limitation however, it has been acknowledged as well as the nonexistence of care professionals.
I just have a few comments to the paper in the current form:
My suggestion is to change keywords because some are very similar, for instance “treatment feedback tool;” and “digital tool;”
Regarding the METHODS and the Expert panel, since the meetings were not recorded, where the researcher notes and report validated by the panel?
Please have a look at the stickers on figures 2-10, they seem unformatted, as well as the subtitles.
Author Response
Responses to the reviewers
Q = question or remark from the editor or reviewer in bold
R = response from the authors
C = changes made in the revised manuscript
Reviewer 1
Dear authors,
First of all, thank you for submitting the manuscript entitled: “I Need to Know: Using the CeHRes roadmap to develop a treatment feedback tool for youngsters with mental health problems” for consideration for publication in the International Journal of Environmental Research and Public Health. According to the information of The Special Issue “The New Normal in Mental Health Care”, authors are invited to submit papers which combine a high academic standard with a practical focus on providing insight into why, how, when, and by whom mental health care can be brought to its new normal. Therefore, the paper is suitable for this Special Issue since it aims to develop a digital PROM feedback tool for use in youth mental health care, a very relevant topic.
The paper is well written and I find the manuscript interesting and very significant in the area, since the use computer technologies, particularly web- applications may help improving the use of self-reported measures, and facilitating youngsters’ active participation in their treatment.
One of the major positive points was the contribution by youngsters in the expert panel, since it might improve the problematical implementation of treatment feedback for youngsters with mental health problems.
As mentioned by the authors, the sample size is a limitation however, it has been acknowledged as well as the nonexistence of care professionals.
R: We appreciate the time and effort of the reviewer to read and comment on our manuscript and thank the reviewer for the feedback and the kind words.
I just have a few comments to the paper in the current form:
Q1: My suggestion is to change keywords because some are very similar, for instance “treatment feedback tool;” and “digital tool;”
R: Thank you for this suggestion. To solve this issue we have chosen to combine related keywords.
C (page 1, Keywords): co-design; youngsters; digital treatment feedback tool; CeHRes roadmap; e-mental Health
Q2: Regarding the METHODS and the Expert panel, since the meetings were not recorded, where the researcher notes and report validated by the panel?
R: In the present study, the meetings were not recorded but the researcher made extensive notes of the meetings. The meeting report did not contain direct quotes, but relevant statements by panel members were paraphrased in the report. The meeting report and the conclusions of the previous meeting have been discussed and validated in the following meeting. We added this information to the description of the expert panel in the Methods section.
C (page 3, section Expert panel): Next, a recap of the previous meeting was given and the report of the previous panel meeting was validated.
Q3: Please have a look at the stickers on figures 2-10, they seem unformatted, as well as the subtitles.
R: Thank you for the suggestion. We uploaded the figures in a separate file.

Reviewer 2 Report
The present study deals with a practical case study on the development of an app for young generation with mental health problems. As the authors point out, recent mobile health apps are not always designed from the user's perspective, and many of them lack usability, especially for the youth - therefore, co-creation by developers, users and companies is highly significant. Based on this recognition, the following points need to be addressed in terms of completeness as an academic paper.
The novelty and progressiveness of this research is unclear. Co-creation by designers and users is practised in many application developments. Why it was not applied in the PROM feedback tool, in partular for mental health-related ones, needs to be mentioned in the Introduction and/or Discussion.
In the description of the Methods, there is a lack of description of the subjects: Any information on the type of disease or its severity ot the youth: When and where were each meeting conducted; For semi-structured/structured interviews, what was the questionnaire like; etc. Above all, the members of this focus group should be more carefully examined to ensure that they adequately reflect the totality of the population and that there is no bias on propensity.
The description of the Result Section is insufficient: The main points of the Phase 1 results should be detailed based on the observed facts obtained; each point of the Phase 2 / Table 2 result should be specifically supported by how many respondents in the interview population. Key findings in the main text should be stated verbatim from the real interview comments, not summarised by the authors.
The following are minor points to be addressed:
- p.1 'a client organisation: which sector? pharma/medical device/IT?
- p.1 'a representative of a client organisation': Which sector? Because it is expected that the desired result will differ depending on the characteristics of the sponsor industry.
- p.10 'give the youngster control about the feedback they want to receive': could they choose what they want to hear only? No more reminders to change their preferences when there was important information once it is defined in the application?
- p.11 'only youngsters themselves can use the INK application': Not even their caregivers? I believe it might be important to share their progress in the treatment with their parents or caregivers especially for mental illness?
- p.15 'Technical and organisational constraints': Could you elaborate more? Due to a client's organisation? or relationship with medical staff?
Finally, the format of the references should be adhered to.
Author Response
Responses to the reviewers
Q = question or remark from the editor or reviewer in bold
R = response from the authors
C = changes made in the revised manuscript
Reviewer 2
Comments and Suggestions for Authors
The present study deals with a practical case study on the development of an app for young generation with mental health problems. As the authors point out, recent mobile health apps are not always designed from the user's perspective, and many of them lack usability, especially for the youth - therefore, co-creation by developers, users and companies is highly significant. Based on this recognition, the following points need to be addressed in terms of completeness as an academic paper.
R: We thank the reviewer for the feedback and also appreciate the time and effort of this reviewer to read and comment on our manuscript.
Q1: The novelty and progressiveness of this research is unclear. Co-creation by designers and users is practised in many application developments. Why it was not applied in the PROM feedback tool, in partular for mental health-related ones, needs to be mentioned in the Introduction and/or Discussion.
R: We thank the reviewer for this remark. Obviously we did not clearly stated the uniqueness of INK. In the Discussion we there added that a feedback tool for youngsters was not available.
C (page 15, Discussion): This study involved the co-design-driven development of INK, a digital PROM-feedback tool for use in youth mental healthcare. Sharing feedback with youngsters in treatment is ubiquitous, but INK is unique. Although there are feedback tools for adults (De Jong et al., 2014; Kelly et al., 2021) or feedback tools that help clinicians to share feed-back with clients (Cooper, Duncan, Golden, & Toth, 2021), feedback tools developed especially for young people in mental healthcare settings in co-creation with youngsters were not available up to now.
Q2: In the description of the Methods, there is a lack of description of the subjects: Any information on the type of disease or its severity ot the youth: When and where were each meeting conducted; For semi-structured/structured interviews, what was the questionnaire like; etc. Above all, the members of this focus group should be more carefully examined to ensure that they adequately reflect the totality of the population and that there is no bias on propensity.
R: The youngsters in the expert panel were all selected based on their experiences with a mental health care institute. We did not collect any background information about the experienced problems or disorders as we were interested in their experiences with questionnaires in treatment. All youngsters had experience with filling out routine outcome questionnaires and not being informed about the results of the questionnaires during treatment. The client organization that participated in this study and through which youngsters were recruited is a large network organization of client advisory boards in the Netherlands. They are especially focussed on youth (mental health) care and provide workshops and guidelines for client advisory boards.
Additional information regarding the semi-structured interviews was added to the paper.
C (page 2, section Expert panel): Seven youngsters aged 12-18 were invited to participate in the panel and all youngsters of them were required to have (previous) experience with youth mental healthcare and to have been treated in a mental health care institute for psychiatric problems. The representative of the client organization recruited the youngsters. The client organization is a network of client advisory boards in care and wellbeing (www.loc.nl). A minimum of three youngsters was required to proceed with individual panel meetings. Every panel meeting was attended by three to seven youngsters (mean attendance: 5.4 youngsters), two user experience-web-designers, a researcher and a representative from a client organization. Panel meetings took place in an informal setting in the region where the youngsters were living on Wednesdays from 17.00-18.30.
C (page 4, Section Formative evaluation): The interpretation of feedback circles was tested using a structured interview. They were shown four different feedback circle options. The one that was most often interpreted in a correct way was chosen for the final version of INK.
Q3:The description of the Result Section is insufficient: The main points of the Phase 1 results should be detailed based on the observed facts obtained;
R: The main point of the Contextual Inquiry executed in Phase 1 was not to obtain some form of insight, but to familiarize the expert panel with the goal of the project and the current situation regarding PROM feedback. We added this in the methods and results section.
C: (page 3, Methods section Phase 1): The main goal in this first phase was to achieve an understanding of future users and the current situation. In the current study, the Contextual Inquiry was initiated by identifying stakeholders and establishing the expert panel to represent future users. In addition, to provide youngsters with insight into the current situation of the PROM feedback that was offered, a patient journey was incorporated (Table 1). This is an interactive method to allow youngsters to become more familiar with the research project and the goal of the co-design-driven development of INK (Simonse, Albayrak, & Starre, 2019).
C: (page 5-6, Results section Phase 1): During the Contextual Inquiry (Phase 1), the trajectory of a typical treatment of youngsters receiving mental healthcare was described using a patient journey. At the start of treatment, youngsters, their parents and sometimes their teachers were asked to fill out questionnaires in a typical treatment trajectory. In the panel meeting the youngsters discussed the order of events in a typical treatment trajectory. For instance, whether they would prefer to complete the questionnaire before attending the mental healthcare institute for the first time or after. The youngsters in the panel meeting said that they only rarely received feedback about the questionnaires they filled out. They would, however, like to be given feedback; preferably directly after they finished the questionnaires.
Q4: each point of the Phase 2 / Table 2 result should be specifically supported by how many respondents in the interview population.
R: The information of Phase 2 and Table 2 was based on all the ten expert panel meetings and the reports that were made of these meetings. Therefore we are unable to clarify how many youngsters indicated that the design requirement was important or unimportant. The reports of specific meetings, however, have been validated in the subsequently meeting.
Q5:Key findings in the main text should be stated verbatim from the real interview comments, not summarised by the authors.
R: Unfortunately, we are unable to provide interview comments verbatim because the interviews and the panel meetings were not recorded. Each statement was based on the validated panel meeting reports.
The following are minor points to be addressed:
- p.1 'a client organisation: which sector? pharma/medical device/IT?
- p.1 'a representative of a client organisation': Which sector? Because it is expected that the desired result will differ depending on the characteristics of the sponsor industry.
R: The client organization is a network organization for client advisory boards in youth care. The representative is a remedial educationalist that supervises several client advisory boards and projects like INK.
- p.10 'give the youngster control about the feedback they want to receive': could they choose what they want to hear only? No more reminders to change their preferences when there was important information once it is defined in the application?
R: That is exactly the case in INK. A youngster can choose to see the information of the questionnaires that is relevant for them. For instance if they not want to see the scores about aggressive behaviour, they do not have to. In this way youngsters are in control and are empowered to choose the topics that are relevant for their treatment.
C (page 10, Results Phase 2 value 4): Several features of INK give the youngster control about the feedback they want to receive. For example, users are able to select or hide the informant or the scale scores they want to see.
- p.11 'only youngsters themselves can use the INK application': Not even their caregivers? I believe it might be important to share their progress in the treatment with their parents or caregivers especially for mental illness?
R: The reviewer is right that also parents and caregivers have to be informed about treatment progress of their child. This is in line with the value conflict which we described in the discussion regarding youngsters and clinicians. In INK, we favour the youngster’s wish to be in control and gain autonomy. The starting point for the development of INK was to provide youngsters with adequate information about their treatment progress. In further research the value conflict has to be addressed by for instance developing a feedback tool for parents that takes into account the preferences of parents.
C (page 18, Discussion section Future research): Improving the flexibility of INK so that it may be used in multiple care settings and by various groups, such as parents, is a third important avenue for future research.
- p.15 'Technical and organisational constraints': Could you elaborate more? Due to a client's organisation? or relationship with medical staff?
R: INK is developed on a limited budget and time. To mark questions and express agreement in INK, safe server storage has to be organised for each client. In addition, to enable reminders and a calendar functionality in INK, the tool has to be more integrated in electronic patient file used by a particular institute. Within the budget, it was unfortunately not possible to make a secure connection between INK and the electronic patient file.
Finally, the format of the references should be adhered to.
R: We checked all references.

Round 2
Reviewer 2 Report
The reviewer confirms that generally appropriate action has been taken to each point of the previous peer review. Congratulations.
Please mention in research limitations the reasons that cannot be fully addressed to the points raised in Q4 and Q5 (and some in the Minor Comments).